# Validation of the Spanish Version of the Double Knowledge Expectations and Received Knowledge Significant Other Scale for Informal Caregivers of People with Dementia (KESO-DEM/RKSO-DEM)

**DOI:** 10.3390/ijerph19095314

**Published:** 2022-04-27

**Authors:** Cindy E. Frías, Claudia Casafont, Esther Cabrera, Adelaida Zabalegui

**Affiliations:** 1Hospital Clinic of Barcelona, 08836 Barcelona, Spain; cfrias@clinic.cat (C.E.F.); casafont@clinic.cat (C.C.); 2School of Health Sciences, TecnoCampus, Pompeu Fabra University, 08302 Mataro, Spain; ecabrera@tecnocampus.cat; 3Department of Care Management and Social Work, Sechenov University, 119435 Moscow, Russia; 4Department of Nursing, Hospital Clinic of Barcelona, 08836 Barcelona, Spain

**Keywords:** dementia, informal caregivers, questionnaire validation, nurse

## Abstract

Dementia is associated with cognitive decline. Becoming an informal caregiver raises questions, requiring information and support from health professionals to guide home care. A multicenter, longitudinal study was carried out to validate the Spanish version of the double scale of expected and received knowledge for informal caregivers of people with dementia (KESO-DEM/RKSO-DEM), the analysis of the dimensional structure of the instrument, its validity and reliability, and temporary stability was carried out. An analysis of criterion and construct validity, internal consistency, and test–retest stability was performed. The evaluation of the interrelation between dimensions was statistically significant. Regarding internal consistency, the scale values were good both for the scale totals and for each dimension of knowledge, with Cronbach’s alpha coefficients of 0.97. For criterion validity, all items showed temporal stability for both questionnaires (*p* < 0.05). The availability of a valid, reliable tool for the measurement of expected and received knowledge in caregivers of people with dementia allows an approach based on the real needs of the family and the patient. It is important to design care protocols for people with dementia that are adapted to their needs and expectations and to their non-curative treatment, to improve the emotional well-being of patients and informal caregivers.

## 1. Introduction

Dementia is associated with cognitive impairment, neuropsychiatric disorders, and changes in mood [1]. Over the years of the course of the illness, the characteristic symptoms of dementia lead to loss of autonomy in the person affected so that a family member becomes an Informal Caregiver (IC). The IC is a family member, or on occasion a friend or neighbor, who assumes the role of main caregiver, taking responsibility for daily care of the people with dementia (PwD), meeting basic and instrumental daily life needs and ensuring the patient remains in a good state of health [1].

However, the chronic nature of the illness increases dependency and, in advanced stages, admission to long-term care centers may become the patient’s main option as the family cannot assume responsibility for care provision [2,3].

When carrying out the IC role, questions and/or a need for information arise, and the support of health professionals is vital as they can guide home care by increasing knowledge and coping skills and lowering the levels of stress produced by caring for someone with a high degree of dependency [2,3].

The approach to the IC role and the situation of the person with dementia (PwD) includes promoting the empowerment of the IC and the PwD as participation in decision-making is fundamental to effective healthcare provision. The creation of empowerment is one of the basic aims in the care process and is achieved through the collaborative relationship between the family and health professionals [2,3,4,5].

In the European Health Futures Forums through the empathy project: empowering patients in the management of chronic diseases, a working group was formed in 2015. The group consisted of organizations and individuals to improve patient empowerment and extend their networks [6], based on the strategic lines of empowerment proposed by the World Health Organization (WHO) [7] as a measure for social promotion, self-help, and self-management of the illness. Thus, the concept of empowerment as a care tool is widely used in nursing with the aim of strengthening the professional–patient relationship and promoting preventive nursing and health education [8,9]. In chronic illnesses such as dementia, the ultimate goal is to help informal caregivers to acquire coping strategies and reduce levels of burden to improve emotional wellbeing [8,9].

The Global Conference on Primary Healthcare in Astana in 2018 declared the aim of focusing a high standard of health in the primary care setting. One of their commitments was to empower individuals and communities in their participation in development of policies and plans [10]. Caregiver education is found to be relevant to the distress and severity of PwD. To improve quality of care, healthcare professionals should provide individualized interventions [11].

Leino-Kilpi et al. (2020), state that empowerment is fundamental in the care process for various health issues and should be one of the care objectives in nursing [12]. They add that the knowledge that facilitates empowerment can be divided into six dimensions: biophysiological, functional, experiential, ethical, social, and financial [13]. In this sense, the patient and her family should receive the information at the right time and in the right context to allow new knowledge to be generated. In dementia, PwD and IC have their own expectations about their need for health education related to the disease; therefore, to satisfy these needs, the empowerment process could follow an appropriate course. In this way, knowledge expectations and knowledge received are important elements to improve the quality of health education for patients and their families.

In the case of dementia, these dimensions can be related to the expected and received knowledge of ICs and PwD to allow illness management at home. As such, all health professionals should assess patient and IC knowledge needs and related perceptions to work on and improve this empowerment [14,15]. Instruments to measure expected and received knowledge have been validated in orthopedic and oncology patients to determine barriers to empowerment by identifying unmet knowledge expectations. Patient and family education involves not only disseminating information but also ensuring that the information is understood and integrated into care provision [15,16]. However, validated instruments that analyze the knowledge expectations of ICs are scarce in the literature. The most extensive and widely used tool in Europe is the double Knowledge Expectations and Received Knowledge Significant Other Scale (KESO/RKSO), which allows assessment of knowledge related to surgical processes and is based on psychosocial theories, and the “Patient Health Engagement” (PHE) model, applied to empowerment of patients through education [16,17,18].

The instrument dimensions allow work on the meaning of the illness, control of one’s own health, and means of support. The biophysiological dimension deals with aspects including the illness, symptoms, and treatment. The cognitive covers knowledge of one’s own health and health problems, and the capacity to obtain, assess, and use this knowledge. The functional addresses the functions of one’s body and mind, including mobility, rest, nutrition, and having the strength and capacity to act with respect to the health issue. The experiential dimension is based on previous experience of empowerment and management, as well as the emotions associated with them. The ethical element is defined as the experience of being valued and respected as an individual, and the feeling that one’s safety is ensured. The social dimension has to do with the capacity for social interaction with, for instance, families, the patient network, and caregivers. The financial element deals with benefits and costs related to health or illness and their connection with self-management [17,19].

Following this line of work, the approach in nursing also permits improvement of the quality of life (QoL) of the PwD and the IC as the non-pharmacological interventions based on psychoeducation, taking knowledge needs and prior experience into account, and significantly improves physical and emotional, educational, and social aspects related to better QoL in patients and families. In turn, these allow changes of attitude to the illness, home care, and decision-making [20].

This underlines the need to develop a modified version of the instrument designed to meet the educational and information requirements of this group, allow in-depth analysis of general and specific aspects of dementia, and be applied to the different stages of the illness.

The KESO/RKSO scale [21] has been validated in contexts within the hospital setting to assess the expectations of knowledge and received knowledge from the patient, with the aim of achieving their empowerment; but at the moment, it is not available as a validated instrument in Spanish that assesses the knowledge expectations and received knowledge of informal caregivers, especially those whose role is focused on responding to the care needs of people with dementia.

During medical and nursing control visits, informal caregivers frequently express doubts related to home care and social support offered by the health system, which highlights the need for formal guidance and the need to develop a modified version of KESO/RKSO scale to respond to educational and information needs and to be able to carry out an in-depth analysis of the general and specific aspects of dementia and to be applied correctly according to the stage of the disease.

The aim of this study was to translate, adapt, validate, and analyze the psychometric properties of the Spanish modified version of the expectations and knowledge received significant other scale in dementia population (KESO-DEM/RKSO-DEM).

## 2. Materials and Methods

### 2.1. Study Design

The study was a multicenter, longitudinal, descriptive study intended to validate the Spanish version of the double expected and received knowledge scale for informal caregivers of people with dementia (KESO-DEM/RKSO-DEM) in 3 care centers specializing in the care of people with cognitive impairment in the Barcelona province: Hospital Clínic of Barcelona, Fundació Sanitària Mollet, and Hospital Pere i Virgili of Barcelona. Recruitment took place between January 2019 and February 2020.

### 2.2. Setting and Participants

The study was carried out in the catchment areas of three centers specializing in the care of people with cognitive impairment in the province of Barcelona and the included participants formed part of the INFOSA-DEM study [22].

The population of interest consisted of family caregivers of people with a diagnosis of dementia treated at one of three centers and living at home with an identified caregiver who does not receive financial compensation for their work. Inclusion criteria were: family caregivers (over 18 years old) of people aged 65 years or older with a diagnosis of dementia and an MMSE test score < 24, and living with the person with dementia or visiting him/her at least twice a week. People with other psychiatric illnesses or Korsakov’s syndrome were excluded.

Administration of questionnaires and signing of informed consent was carried out at the patient’s home or designated center according to availability. To assess the temporal stability of the double scale, the visit was repeated and the questionnaire re-administered 15 days after baseline.

The study was conducted in two phases. During the first phase, the modified Spanish version of the double expected and received knowledge scale of caregivers of people with dementia was produced (KESO-DEM/RKSO-DEM), analyzing conceptual and semantic equivalence and item content validity of the generated version through a pilot test and assessment by a multidisciplinary group of experts. In the second phase, using confirmatory procedures with the whole selected sample, the instrument’s dimensional structure, validity and reliability, and temporal stability were analyzed. Methodological standards recommended in International Test commission guidelines (2018) from the guide created by Sousa and Wilaiporn (2011) were followed in the translation and back-translation of the instrument. Streinner and Kottner recommendations were also followed [22,23,24,25].

### 2.3. Data Collection/Instrument

Measurements collected include sociodemographic data of PwD and informal caregivers, onset and type of dementia, cognitive status with Mini-Mental State Examination (MMSE) [26,27], and comorbidity (Charlos Index) [28].

The KESO/RKSO questionnaire [16,17] is a double scale that allows assessment of the expected and received knowledge of informal caregivers. Each scale consists of 40 items divided into 6 dimensions or knowledge categories: biophysiological (8 items on the illness, symptoms, treatments and complications), functional (8 items on nutrition, rest and body hygiene), experiential (3 items related to experiences and feelings), ethical (9 items that address participation in decision-making, confidentiality, rights, and responsibilities), social (6 items on caregivers, support personnel and patient organizations), and financial (6 items on costs). The response format is a Likert-type scale with the options: 0 = not applicable in my case; 1 = fully disagree; 4 = fully agree. High scores indicate greater expectations or more knowledge received by families. Permission was sought from the author to adapt the instrument prior to the commencement of the study.

### 2.4. Translation and Back-Translation of the Original Instrument

Specified directives were followed for the translation and back-translation of the instrument [24,29]. Two bilingual, bicultural translators with wide experience in the field of health were chosen to separately and independently translate the instrument from English to Spanish. The two versions were compared with the original by the members of the research team to resolve any ambiguities and discrepancies. This preliminary version was translated again (back-translation) by another two translators working independently to create two back-translated versions of the instrument in its original language. The members of the research team compared the two back-translations with each other and with the original instrument with respect to similarity of instructions, items, drafting of response format, semantic structure, meaning, and relevance. From this process, and once consensus on grammatical and cultural aspects had been reached, some modifications were made.

Through a review of the literature and based on the knowledge categories and clinical experience in the study pathology, the research team added a total of six of their own statements specific to dementia to the original instrument matrix. These statements were validated by informal caregivers and an expert panel on dementia and caregiving. The same format and writing style were maintained for the statements, resulting in a prefinal version of the KESO-DEM/RKSO-DEM instrument in Spanish, consisting of 46 items. Adding these items completed the approach for dementia care where cognitive function, availability of resources and legal representation are main elements to be considered. The prefinal version of the KESO-DEM/RKSO-DEM can be found in Appendix A.

### 2.5. Psychometric Properties

Assessment of the psychometric properties of the KESO-DEM/RKSO-DEM double scale included analysis of content, construct and criterion validity, and instrument reliability.

Conceptual equivalence and semantic clarity of the generated version was analyzed using the debriefing technique [30] in a group of experts and was pilot tested in 30 informal caregivers from the study population. Each participant was asked to complete the KESO-DEM/RKSO-DEM questionnaire and to respond to six dichotomous questions with the aim of analyzing the response format, instructions, and statements. Information on the time taken to fill out the questionnaire was also collected. Any items considered confusing by at least 20% of the subjects were reassessed by the team [30]. In parallel, the readability of the instrument statements was analyzed using the INFLESZ tool [31].

Content validity was assessed by a multidisciplinary panel of experts consisting of physicians, nurses, and social workers through calculation of the content validity index for items (I-CVI) and for the scale (S-CVI). The members assessed each item according to its relevance through a Likert-type scale with four response options (1 = not relevant; 2 = relevance cannot be assessed; 3 = relevant but requiring minor modification; 4 = highly relevant). Statements in 1 and 2 were reviewed until consensus was reached. Figure 1 illustrates the translation, adaptation, and content validity process applied to the Spanish version of the modified KESO-DEM/RKSO-DEM scale.

To analyze construct validity, an exploratory factor analysis was performed using varimax rotation to confirm the six dimensions that made up the original instrument. A value of >1 was established for the determination of the categories and the suitability of the sample was analyzed using the Bartlett and Káiser–Meyer–Olkin (KMO) tests. Criterion validity was explored through the correlation between the KESO-DEM dimensions and the Family Inventory of Needs (FIN) [32], which assesses family care needs and consists of two subscales: one determining the importance of the care need and the other satisfaction with the care cover required. It consists of 13 items scored on a Likert-type scale where 0 represents “not at all” or “not covered” and 4 represents “completely covered”. The scores are calculated separately for each subscale, and the RKSO-DEM with the Preparedness for Caregiving Scale (PCS) [33] which assesses the state of caregivers’ perceived preparation to provide care. It consists of 8 items with a Likert-type response format where 0 represents “not being prepared” and 4 “feels very well prepared”. Score totals range from 0 to 32 points. A low score demonstrates a greater need to be prepared. Spearman correlation coefficients were evaluated to determine the intercorrelation between the scale dimensions.

To analyze reliability, internal consistency was calculated for the KESO-DEM/RKSO-DEM scale and for each of the dimensions using Cronbach’s alpha, with values > 0.7 considered acceptable. Temporal stability was analyzed using the intraclass correlation coefficient [34] between the questionnaires administered at the baseline visit and at 15 days.

For study purposes, questionnaires with responses to at least 50% of the items were included. Data were analyzed using the statistical software package R V.3.2.3 for Windows.

## 3. Results

### 3.1. Participants

Initially, 628 subjects were eligible to participate in the study and, finally, 471 caregivers were contacted by telephone. Finally, 159 informal caregivers of people with dementia who met inclusion criteria were included. The median age was 68.6 (SD13.4) and 52.2% were women. The most prevalent type of dementia was Alzheimer’s (Table 1).

### 3.2. Validity

Of the 30 caregivers, response options appeared to be clear for 93%, with 100% of items found to be of interest. The length of the questionnaire was considered adequate by 90% of the informal caregivers. From the clarity and legibility of statements analysis, using the INFLESZ tool, scores of 70.1 (very easy) were obtained for both questionnaires and the mean response time for questionnaire completion was 16 min. For the items, exploration of content equivalence through calculation of the content validity index (I-CVI = 0.78) and of the scale (S-CVI = 0.847), and by calculating the average S-CVA/Ave = 0.98 (>0.95), values above the minimum acceptable were obtained (Table 2).

### 3.3. Reliability

Analysis was conducted on construct and criterion validity, internal consistency, and test–retest stability of the new adapted version of the KESO-DEM/RKSO-DEM instrument in the 161 participants. The factor analysis with varimax rotation showed seven factors with a value > one for the KESO-DEM scale, and six factors with a value > one for the RKSO-DEM (Table 3). Assessment of the interrelationship between dimensions was performed using the Spearman correlation coefficient, with KESO-DEM (r = 0.46–0.79) and RKSO-DEM (r = 0.47–0.79) as statistically significant (Table 4).

Regarding the internal consistency of the instrument, the scale values were good for both the scale totals and each knowledge dimension, with Cronbach’s alpha coefficients of 0.97 for the KESO-DEM and 0.98 for the RKSO-DEM. For criterion validity, the FIN and the PCS were taken as reference, showing statistically significant Spearman correlation coefficients of *p* < 0.05 in all the dimensions and in the scale totals for the FIN (A) and PCS (Table 5). Test–retest stability was used to determine the reliability of the scores over time. Table 6 shows the intraclass correlation coefficients (ICC) for items on both scales. The instrument (retest) was completed by 112 caregivers 15 days after the first administration. All items showed temporal stability for both questionnaires (*p* < 0.05), with values between 0.396 and 0.804 (Table 6).

## 4. Discussion

This study highlights the importance of designing care protocols for people with dementia living at home that are adapted to their needs and expectations and to their non-curative treatment, whose interventions should be addressed to improve their emotional wellbeing and the QoL of both patients and informal caregivers [20,22]. Thus, it is vital to have validated instruments available that assess the information needs of caregivers and promote the identification of real needs so that training activities can be developed and home management of the illness can be improved [1]. The results obtained in this study underline the good validity and reliability of the double KESO-DEM/RKSO-DEM scale, which has been shown to be a tool that facilitates analysis of expected and received knowledge in caregivers of people with dementia, based on meaningful analysis of its psychometric properties and stability, as well as its ease of comprehension.

The factor analysis of the data obtained supports the construct validity of the adapted Spanish version of the instrument, confirming the six dimensions of the original scale for the RKSO-DEM and the seven dimensions for the KESO-DEM. Structured criteria were followed to assess the specificity of the instrument and its subscales to measure expected and received knowledge, semantics, and clarity. Results are similar to those found in the validation of the instrument in other populations and contexts [17].

Criterion validity of the instrument showed a positive relationship between knowledge expectations and the family’s need for information. These results confirm that the greater the need for information, the more knowledge the caregivers expect to receive. As such, education of patients and caregivers as a nursing goal is fundamental in achieving positive health outcomes and cost-effectiveness [12,13]. In fact, various studies indicate that some caregivers experience certain difficulties obtaining information on subjects related to their family member and the illness. This results in worse functioning in their role as caregivers, as a central need is good quality information in order to make the best health decisions [35,36]. Our results show that regarding knowledge received and preparation for care, a positive relationship was also observed, demonstrating that caregivers who have received more knowledge are better prepared to care for the PwD, especially at home [37]. They have enough self-confidence to control illness-related symptoms at home, correctly administer the prescribed treatment to their relative, and follow the recommendations of health professionals [38]. In caregivers of PwD, this translates into empowerment, which leads to improvement in their capacities and reduction in their limitations for care, so improving their quality of life [39,40]. Advanced Practice Nurses in the Primary care setting are key in establishing people-centered integrated care, especially for older people. Additionally, inter-sectorial collaboration improves continuity of care [41].

Estimated internal consistency was high for both scales, with values in line with those reported for the original instrument; Cronbach’s alpha 0.98 for the KESO and 0.99 for the RKSO [42], measured through the intraclass correlation coefficient, which showed stable scores over time. The availability of a valid, reliable tool for the measurement of expected and received knowledge in caregivers of people with dementia allows an approach based on the real needs of the family and which is adapted to the patient’s stage of illness. Therefore, it can be used in Spanish-speaking countries and can serve as a basis for the design of instruments that permit measurement and work on the empowerment of patients and caregivers, with education by nurses as a facilitating process in healthcare.

This study has some limitations. First, participants in this study were only enrolled from three centers in the province of Barcelona. Second, the sample could not be randomized due to the difficulty of recruiting ICs, as there was no specific list of caregivers of PwD. Third, although a calculation of the sample size was made for this validation study, due to a lack of consensus and clear guidelines on sample size calculation for validation studies [43], the sample size for the INFOSA-DEM [22] project was considered, in which it was determined that 160 participants would need to be included to study the effectiveness of a psychoeducational intervention for informal caregivers of people with dementia.

## 5. Conclusions

The KESO-DEM/RKSO-DEM scale it is a valid and reliable instrument for measuring expectations of knowledge and knowledge received by informal caregivers of people with dementia, with good psychometric properties. This instrument adds value to the care provided by informal caregivers of PwD and can identify their expected and received knowledge. Therefore, this instrument could be used to develop future guidelines to empower informal caregivers of PwD.

The findings could influence future nursing practice, research, and leadership. The scale allows professionals to identify and focus on which areas professional guidance in health should be deepened in informal caregivers of people with dementia, this approach being orientation to guide the home care of other pathologies.

## Figures and Tables

**Figure 1 ijerph-19-05314-f001:**
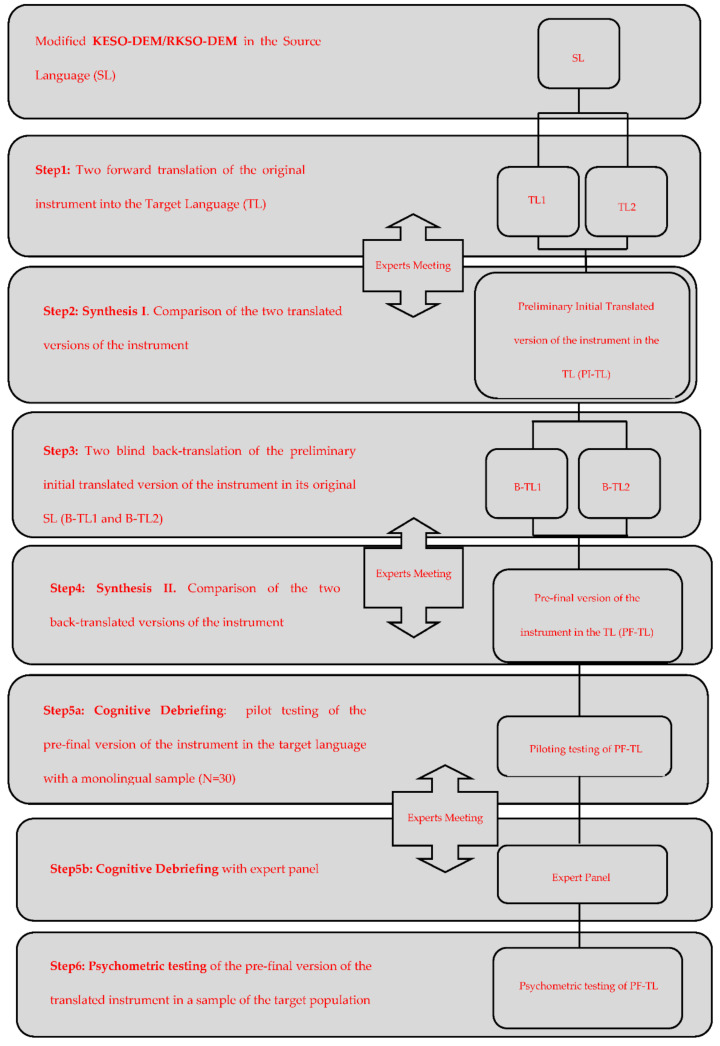
Translation, adaptation, and content validity process of the Spanish version of the KESO-DEM/RKSO-DEM scale.

**Table 1 ijerph-19-05314-t001:** Sample characteristics (*n* = 159).

Variable	*n* (%)
Age of caregiver, years	68.6 ± 13.4
Age of patient, years	78.9 ± 8.2
Patient gender, female	83 (52.2%)
Type of dementia	
Alzheimer’s	103
Cerebrovascular	1
Vascular	11
Lewy	6
Unknown	16
Other	4
Onset of dementia	128/151 (84.8%)
Time since diagnosis, years	5.3 ± 3.2
Cognitive state (MMSE)	16.5 ± 7.6
Normal (27–30)	3/89 (3.4%)
Low severity (21–26)	34/89 (38.2%)
Moderate severity (15–20)	24/89 (27.0%)
High severity (<15)	28/89 (31.4%)
Comorbidity (Charlson Index)	1.4 ± 1.2

**Table 2 ijerph-19-05314-t002:** Content validity.

Item	E1	E2	E3	E4	E5	E6	E7	E8	E9	E10	I-CVI Pre/Post
Biophysiological											
1.1. Symptoms related to the illness	4	4	4	4	4	3	3	4	4	4	1
1.2. When contact was made with the hospital	4	4	3	4	4	3	3	4	3	4	1
1.3. Tests to be carried out	3	4	4	4	4	3	4	3	3	4	1
1.4. How one should prepare for the tests	3	4	4	4	4	3	3	3	4	4	1
1.5. Obtain test results	3	4	4	4	3	3	3	3	3	4	1
1.6. The different treatments available	4	4	4	4	4	3	4	4	3	4	1
1.7. Complications related to treatment	4	4	4	4	4	3	4	4	4	4	1
1.8. Prevent complications	3	4	4	4	4	3	4	4	4	4	1
Functional											
2.9. Ensure that my needs are covered	3	4	4	4	4	3	4	4	4	3	1
2.10. What types of physical exercise I can do	4	3	4	4	3	3	3	4	3	3	1
2.11. How much do I need to rest	4	4	4	4	3	3	4	4	3	3	1
2.12. What type of diet is most suitable	4	4	4	4	4	3	3	4	3	3	1
2.13. How to use the bathroom (e.g., shower, have a bath)	4	4	4	4	4	3	3	4	3	3	1
2.14. How illness/treatment affects bodily functions	3	4	4	4	4	3	3	4	3	3	1
2.15. How illness/treatment affects cognitive functions	4	4	4	4	4	3	4	4	4	4	1
2.16. How to manage possible cognitive alterations	4	4	3	4	4	3	4	4	4	4	1
2.17. How illness/treatment affects home organization	3	4	4	4	4	3	4	4	4	4	1
2.18. Where to obtain the help I need for care	4	4	4	4	4	3	4	4	3	4	1
Experiential											
3.19. Feelings caused illness/treatment of the person receiving care	4	4	4	4	4	3	4	4	4	4	1
3.20. Feelings causedillness/treatment to the person receiving care	3	4	4	4	4	3	4	4	4	3	1
3.21. Who to talk to about feelings caused by illness/treatment	3	3	4	4	4	3	4	4	4	3	1
3.22. How to take advantage of previous experiences	2/4	3	4	4	3	3	4	4	3	3	0.9/1
Ethical											
4.23. Person receiving care participates in decision-making	4	4	4	4	4	3	4	4	3	4	1
4.24. Person receiving care can express own opinion and point of view	3	4	4	4	4	3	4	4	3	4	1
4.25. Rights of the person receiving care	3	4	4	4	4	3	4	4	3	3	1
4.26. Responsibility regardingcare	4	3	3	4	4	3	4	4	3	3	1
4.27. The patient representative and their work	2/4	2/4	1/4	3	4	3	4	2/1	3	3	0.6/0.9
4.28. Grant the power to be represented	4	4	4	4	4	3	4	3	4	4	1
4.29. Responsibilities of the different care professionals	4	4	4	4	4	3	4	4	3	4	1
4.30. Confidentiality of clinical history data	1/4	4	4	4	4	3	4	4/1	3	3	0.9
4.31. Who has access to clinical history	1/4	4	4	4	4	3	4	4	3	2/3	0.8/1
4.32. Obtain access to clinical history	1/4	4	4	3	4	3	4	4/2	3	2/4	0.8/0.9
Social											
5.33. Who informs about illness/treatment issues	4	4	4	4	4	3	4	4	3	4	1
5.34. Participate in care	4	4	4	4	4	3	3	4	4	4	1
5.35. Connect care to social life and hobbies	3	4	4	4	4	3	3	3	4	4	1
5.36. Involve family and/or others in the environment	3	4	4	4	4	3	3	4	4	4	1
5.37. Obtain a support person if needed	4	4	4	4	4	3	3	4	4	4	1
5.38. Obtain more care or treatment if necessary	4	3	4	4	4	3	3	3	4	4	1
5.39. Contact the priest	1/4	2/4	1/4	3	4	3	3	3	3	1/2	0.6/0.9
5.40. Patient organizations and their activities	3	4	4	4	4	3	3	3	3	4	1
Economy											
6.41. Care and itscosts	2/4	4	4	4	4	3	3	3	3	4	0.9/1
6.42. Obtain help due to the illness	4	4	4	4	4	3	3	3	3	4	1
6.43. Insurance and cover for treatment	1/4	2/4	3	4	4	3	2	3	3	3	0.7/0.9
6.44. Rehabilitation and adaptationcourses	1/4	4	3	4	4	2	3	3	3	4	0.8/0.9
6.45. Home care and nursing home costs	4	4	4	4	4	2	3	4	3	4	0.9
6.46. Medication costs	3	4	4	4	4	3	3	4	3	4	1
I-CVI: Item content validity index											

In the case that there are modifications in pre- and post-scores, this should be indicated with separation of the two with “/”.

**Table 3 ijerph-19-05314-t003:** Main components analysis with varimax rotation (Reliability).

Item	Information Expected by Informal Caregivers (EKSO-DEM)								Knowledge Received by Informal Caregivers (RKSO-DEM)						
	RC2	RC1	RC6	RC4	RC5	RC7	RC8	RC3	RC4	RC1	RC5	RC2	RC6	RC3	RC7
	Biophysiological	Functional1	Functional2	Experiential	Ethical	Social1	Social2	Economic	Biophysiological	Functional	Experiential/Ethical	Social1/Economic	Social2		
Biophysiological															
1.1. Symptoms related to the illness	0.81									0.48					
1.2. When contact was made with the hospital	0.67								0.74						
1.3. Tests to be carried out	0.85								0.50						
1.4. How one should prepare for the tests	0.82								0.47						
1.5. Obtain test results	0.83											0.62			
1.6. The different treatments available	0.82								0.58						
1.7. Complications related to treatment	0.77								0.63						
1.8. Prevent complications	0.78								0.56						
Functional															
2.9. Ensure that my needs are covered		0.44								0.79					
2.10. What types of physical exercise I can do		0.53								0.55					
2.11. How much do I need to rest		0.41								0.75					
2.12. What type of diet is most suitable		0.42								0.74					
2.13. How to use the bathroom (e.g., shower, have a bath)		0.62								0.70					
2.14. How illness/treatment affects bodily functions			0.62							0.64					
2.15. How illness/treatment affects cognitive functions			0.72							0.70					
2.16. How to manage possible cognitive alterations			0.72							0.76					
2.17. How illness/treatment affects home organization		0.49								0.75					
2.18. Where to obtain the help I need for care				0.44								0.52			
Experiential															
3.19. Feelings caused illness/treatment of the person receiving care				0.73							0.60				
3.20. Feelings caused illness/treatment to the person receiving care				0.75							0.46				
3.21. Who to talk to about feelings caused by illness/treatment				0.67							0.47				
3.22. How to take advantage of previous experiences						0.63					0.59				
Ethical															
4.23. Person receiving care participates in decision-making					0.59						0.76				
4.24. Person receiving care can expressown opinion and point of view					0.54						0.76				
4.25. Rights of the person receiving care							0.54				0.69				
4.26. Responsibility regardingcare					0.43						0.57				
4.27. The patient representative and their work					0.54						0.50				
4.28. Grant the power to be represented					0.60						0.77				
4.29. Responsibilities of different care professionals					0.66						0.40				
4.30. Confidentiality of clinical history data					0.80						0.48				
4.31. Who has access to clinical history					0.84						0.49				
4.32. Obtain access to clinical history					0.74									0.72	
Social															
5.33. Who informs about illness/treatment issues						0.53						0.76			
5.34. Participate in care							0.41						0.43		
5.35. Connect care with social life and hobbies							0.48						0.41		
5.36. Involve families and/or people in the neighborhood						0.69							0.65		
5.37. Obtain a support person if needed								0.75				0.71			
5.38. Obtain more care or treatment if necessary								0.79				0.62			
5.39. Contact the priest						0.47								0.61	
5.40. Patient organizations and their activities						0.56					0.47				
Economic															
6.41. Care and its costs								0.43						0.61	
6.42. Obtain help due to the illness								0.87				0.79			
6.43. Insurance and cover for treatment								0.44			0.51				
6.44. Rehabilitation and adaptationcourses								0.61				0.67			
6.45. Home care and nursing home costs								0.61				0.66			
6.46. Medication costs								0.41				0.49			

**Table 4 ijerph-19-05314-t004:** Intercorrelations between dimensions.

EKSO-DEM Dimensions	Biophysiological	Functional	Experiential	Ethical	Social
Functional	0.697				
Experiential	0.501	0.672			
Ethical	0.483	0.667	0.742		
Social	0.466	0.698	0.690	0.747	
Economic	0.474	0.645	0.596	0.748	0.797
**RKSO-DEM Dimensions**	**Biophysiological**	**Functional**	**Experiential**	**Ethical**	**Social**
Functional	0.705				
Experiential	0.519	0.742			
Ethical	0.564	0.664	0.680		
Social	0.677	0.716	0.646	0.639	
Economic	0.646	0.606	0.491	0.479	0.773

All Spearman correlation coefficients are statistically significant at a significance level of 0.05.

**Table 5 ijerph-19-05314-t005:** Internal consistency and external validity.

Dimensions	Internal Consistency		External Validity	
	Mean (SD)	Cronbach’s Alpha	Needs Inventory (A)	Needs Inventory (B)
EKSO-DEM				
Biophysiological	3.72 (0.50)	0.94	0.083	−0.096
Functional	3.68 (0.51)	0.92	0.216 *	−0.103
Experiential	3.57 (0.59)	0.83	0.297 *	−0.118
Ethical	3.37 (0.66)	0.92	0.395 *	−0.100
Social	3.38 (0.63)	0.87	0.398 *	−0.030
Economic	3.37 (0.71)	0.87	0.330 *	−0.058
Scale total	3.52 (0.49)	0.97	0.375 *	−0.077
RKSO-DEM				
Biophysiological	2.21 (0.82)	0.89	−0.192 *	0.462 *
Functional	1.76 (0.81)	0.94	−0.065	0.432 *
Experiential	1.51 (0.75)	0.89	0.046	0.374 *
Ethical	1.46 (0.63)	0.91	0.003	0.350 *
Social	1.68 (0.64)	0.85	−0.154	0.304 *
Economic	1.76 (0.70)	0.82	−0.195 *	0.345 *
Scale total	1.74 (0.62)	0.97	−0.134	0.457 *

* Spearman correlation coefficients statistically significant at a significance level of 0.05. SD: Standard deviation.

**Table 6 ijerph-19-05314-t006:** Test–retest temporal stability (reproducibility). All items have temporal stability (intraclass correlation coefficient *p* value lower than 0.05).

Item	EKSO-DEM	RKSO-DEM
Biophysiological	0.676	0.727
1.1. Symptoms related to the illness	0.616	0.656
1.2. When to contact the hospital	0.612	0.771
1.3. Tests to be carried out	0.706	0.629
1.4. How one should prepare for the tests	0.689	0.673
1.5. Obtain the test results	0.664	0.660
1.6. The different treatments available	0.607	0.626
1.7. Complications related to treatment	0.442	0.671
1.8. Preventcomplications	0.576	0.712
Functional	0.748	0.776
2.9. Ensure that my needs are covered	0.645	0.701
2.10. What types of physical exercise I can do	0.622	0.699
2.11. How much do I need to rest	0.836	0.757
2.12. What type of diet is most suitable	0.608	0.690
2.13. How to use the bathroom (e.g., shower, have a bath)	0.624	0.743
2.14. How the illness/treatment affects bodily functions	0.657	0.680
2.15. How illness/treatment affects cognitive functions	0.652	0.761
2.16. How to manage possible cognitive alterations	0.752	0.740
2.17. How illness/treatment affects home organization	0.673	0.732
2.18. Where to obtain the help I need for care	0.604	0.677
Experiential	0.753	0.682
3.19. Feelings caused illness/treatment of the person receiving care	0.711	0.609
3.20. Feelings caused illness/treatmentto the person receiving care	0.702	0.692
3.21. Who to talk to about feelings caused by illness/treatment	0.681	0.664
3.22. How to take advantage of previous experiences	0.829	0.712
Ethical	0.744	0.594
4.23. Person receiving care participates in decision-making	0.754	0.672
4.24. Person receiving care can express own opinion and point of view	0.732	0.639
4.25. Rights of the person receiving care	0.604	0.533
4.26. Responsibility regardingcare	0.632	0.607
4.27. The patient representative and their work	0.543	0.695
4.28. Grant the power to be represented	0.654	0.609
4.29. Responsibilities of the different care professionals	0.715	0.655
4.30. Confidentiality of clinical history data	0.646	0.563
4.31. Who has access to clinical history	0.639	0.525
4.32. Obtain access to clinical history	0.716	0.566
Social	0.648	0.654
5.33. Who informs about illness/treatment issues	0.640	0.634
5.34. Participate in care	0.580	0.656
5.35. Connect care to social life and hobbies	0.590	0.690
5.36. Involve family and/or people in the neighborhood	0.669	0.610
5.37. Obtain a support person if needed	0.429	0.628
5.38. Obtain more care or treatment if necessary	0.391	0.652
5.39. Contact the priest	0.667	0.348
5.40. Patient organizations and their activities	0.676	0.613
Economic	0.656	0.556
6.41. Care and its costs	0.626	0.455
6.42. Obtain help due to the illness	0.396	0.591
6.43. Insurance and cover for treatment	0.661	0.496
6.44. Rehabilitation and adaptation courses	0.465	0.663
6.45. Home care and nursing home care	0.601	0.643
6.46. Medication costs	0.647	0.514

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
