# Peer review of "Validation of the Spanish Version of the Double Knowledge Expectations and Received Knowledge Significant Other Scale for Informal Caregivers of People with Dementia (KESO-DEM/RKSO-DEM)"

_ijerph, 2022, doi:10.3390/ijerph19095314_

Round 1
Reviewer 1 Report
Thank the authors for the submission of the manuscript entitled “Validation of the Spanish version of the double Knowledge Expectations and Received Knowledge Significant Other Scale for Informal Caregivers of People with Dementia (KSO-DEM/RKSO-DEM)”. The study aimed to translate, adapt, validate and analyze the psychometric properties of the Spanish modified version of the Expectations and knowledge received significant other scale in dementia population (KESO-DEM/RKSO-DEM). Although the some novel ideas are presented, I give some specific comments for authors to improve the manuscript.
It seems this research focus on a local issue in Spain. However, for readers of international journals, the underlying theoretical contribution need to be further highlighted.
In 2.4, authors mentioned “through a review of the literature and based on the knowledge categories and clinical experience in the study pathology, the research team added a total of six of their own statements specific to dementia to the original instrument matrix”. It requires more literature to make a strong argument for six of their own statements.
The procedure of the proposed method should be provided in a flowchart or diagram. The presentation of the results showed some fragility. How about presenting the results obtained on each step in the research on a board?
The pre-final version of the EKSO-DEM/RKSO-DEM instrument in Spanish, consisting of 46 items, should be provided in the appendix.
Author Response
Validation of the Spanish version of the double Knowledge Expectations and Received Knowledge Significant Other Scale for Informal Caregivers of People with Dementia (KESO-DEM/RKSO-DEM).
Barcelona (Spain), March, 23, 2022
Dear Professor Nora Shang,
Managing Editor, MDPI, and anonymous reviewer,
After conducting a careful review of our proposed article, based on all reviewer suggestions, we have submitted them for a new evaluation. In the new manuscript, we have highlighted in red the modifications made to the original text.
We want to express our sincere thanks to the editor and the reviewer for their great work; their notes have allowed us not only to significantly improve the manuscript but also to reflect on future research. Following this letter we detail how we have responded to the suggestions of the editor and the reviewer in the new version of our proposed article.
We hope that the work done will achieve the final approval of the IJERPH Editorial Team. If this is not the case, all authors are available to resolve any issue or proceed with new revisions as necessary.
Respectfully,
The authors.
*************************************
REVIEWER’S COMMENTS
REVIEWER 1
- In 2.4, authors mentioned “through a review of the literature and based on the knowledge categories and clinical experience in the study pathology, the research team added a total of six of their own statements specific to dementia to the original instrument matrix”. It requires more literature to make a strong argument for six of their own statements.
Response 1: We appreciate this evaluation. We have added more information based on the literature in order to follow their recommendations.
- The procedure of the proposed method should be provided in a flowchart or diagram. The presentation of the results showed some fragility. How about presenting the results obtained on each step in the research on a board?
Response 2: Totally agree with your suggestion. From this, we have added a diagram in which the stages are specified
- The pre-final version of the EKSO-DEM/RKSO-DEM instrument in Spanish, consisting of 46 items, should be provided in the appendix.
Response 3: Thank you very much for your suggestion. We have added what you requested.
Reviewer 2 Report
Dear Authors,
This is a Spanish translation of the scale, which I think is very interesting, and I thought the translation procedure was very carefully followed. However, there was no table in the manuscript, and unfortunately, as a reviewer, I could not make a decision that it was acceptable for publication.
Even if there had been a table, there were serious problems in the following sections.
- There were many English sentences that were difficult to understand, such as lines 40-42. Please check the logic and ask a professional proofreader to edit it appropriately.
- It was not clear why you translated this scale into Spanish, i.e., why you want/should use it in Spain. You should state the significance of translating it into Spanish for use. It is insufficient to say that you created it because there is no Spanish version.
- In the text, you state that IC (please define) and PwD empowerment are necessary and that knowledge that promotes empowerment can be divided into six dimensions. It then says, " The knowledge expectations and knowledge received are important elements in improving the quality of health education for patients and the families." This is an important part, but I found it confusing because it is not logically connected. In other words, why is the measurement of expected and received knowledge so important for improving the quality of health education for patients and their families? I was left wondering because the mechanism is not described. The explanation needed to be logical.
Author Response
Validation of the Spanish version of the double Knowledge Expectations and Received Knowledge Significant Other Scale for Informal Caregivers of People with Dementia (KESO-DEM/RKSO-DEM).
Barcelona (Spain), March, 23, 2022
Dear Professor Nora Shang,
Managing Editor, MDPI, and anonymous reviewer,
After conducting a careful review of our proposed article, based on all reviewer suggestions, we have submitted them for a new evaluation. In the new manuscript, we have highlighted in red the modifications made to the original text.
We want to express our sincere thanks to the editor and the reviewer for their great work; their notes have allowed us not only to significantly improve the manuscript but also to reflect on future research. Following this letter we detail how we have responded to the suggestions of the editor and the reviewer in the new version of our proposed article.
We hope that the work done will achieve the final approval of the IJERPH Editorial Team. If this is not the case, all authors are available to resolve any issue or proceed with new revisions as necessary.
Respectfully,
The authors.
*************************************
REVIEWER’S COMMENTS
REVIEWER 2
- There were many English sentences that were difficult to understand, such as lines 40-42. Please check the logic and ask a professional proofreader to edit it appropriately.
Response 1: Thanks for your suggestion. We have changed the paragraph according to your recommendation.
- It was not clear why you translated this scale into Spanish, i.e., why you want/should use it in Spain. You should state the significance of translating it into Spanish for use. It is insufficient to say that you created it because there is no Spanish version
Response 2: Thank you for your comments. Taking into account your comment and the comments of the other reviewers, we have added more information about it.
- In the text, you state that IC (please define) and PwD empowerment are necessary and that knowledge that promotes empowerment can be divided into six dimensions. It then says, " The knowledge expectations and knowledge received are important elements in improving the quality of health education for patients and the families." This is an important part, but I found it confusing because it is not logically connected. In other words, why is the measurement of expected and received knowledge so important for improving the quality of health education for patients and their families? I was left wondering because the mechanism is not described. The explanation needed to be logical.
Response 3: Thanks for your appreciation. We have improved the explanation so that it is understood in a more logical way.
Reviewer 3 Report
General comments
The Ms entitled “Validation of the Spanish version of the double Knowledge Expectations and Received Knowledge Significant Other Scale for Informal Caregivers of People with Dementia (KSO-DEM/RKSO-DEM)” reports about the multicenter, longitudinal study to validate the Spanish version of the double scale of expected and received knowledge for informal caregivers of people with dementia (KESO-DEM/RKSO-DEM). Criterion and construct validity, internal consistency and test-retest stability are the psychometric properties assessed, proving very good internal consistency of the scale. The validation of a scale in a new setting is useful to add novel tools to the armamentarium available to provide patients affected by dementia with the best assistance.
Major comments
The important issue that is addressed in this paper is that all health professionals should assess patient and informal caregiver knowledge needs and perceptions to improve empowerment, ensuring that the information is understood and included into practice. The dimensions of the tool are explained in the Introduction. However, more description about the psychometric properties of the originally developed scale should be included. I suggest the inclusion of a diagram reporting the inclusion and exclusion of participants in the Results, in order to better explain the process. Although the study desing and the statistical analysis are correct, more detail should be applied to the results including the documents produced to perform the translation and the cross-cultural adaptation and the tables for the several aspects of validity, internal consistency and reliability analyzed. The discussion is clear and reports also the study limitations.
Minor comments
Line 306 ” The EKSO-DEM/RKSO-DEM scale it is a valid…”: “it “ should be deleted.
There are some typos.
Author Response
Validation of the Spanish version of the double Knowledge Expectations and Received Knowledge Significant Other Scale for Informal Caregivers of People with Dementia (KESO-DEM/RKSO-DEM).
Barcelona (Spain), March, 23, 2022
Dear Professor Nora Shang,
Managing Editor, MDPI, and anonymous reviewer,
After conducting a careful review of our proposed article, based on all reviewer suggestions, we have submitted them for a new evaluation. In the new manuscript, we have highlighted in red the modifications made to the original text.
We want to express our sincere thanks to the editor and the reviewer for their great work; their notes have allowed us not only to significantly improve the manuscript but also to reflect on future research. Following this letter we detail how we have responded to the suggestions of the editor and the reviewer in the new version of our proposed article.
We hope that the work done will achieve the final approval of the IJERPH Editorial Team. If this is not the case, all authors are available to resolve any issue or proceed with new revisions as necessary.
Respectfully,
The authors.
REVIEWER 3
- The important issue that is addressed in this paper is that all health professionals should assess patient and informal caregiver knowledge needs and perceptions to improve empowerment, ensuring that the information is understood and included into practice. The dimensions of the tool are explained in the Introduction. However, more description about the psychometric properties of the originally developed scale should be included. I suggest the inclusion of a diagram reporting the inclusion and exclusion of participants in the Results, in order to better explain the process.
Response 1: Thank you for your comments. Taking into account your comment and the comments of the other reviewers, we have added more information andhas been modified as suggested.
- Although the study design and the statistical analysis are correct, more detail should be applied to the results including the documents produced to perform the translation and the cross-cultural adaptation and the tables for the several aspects of validity, internal consistency and reliability analyzed. The discussion is clear and reports also the study limitations
Response 2: Thank you for your observation. We had provided this information as attachments to other documents that apparently were not provided to reviewers due to our error. We have already added them to the manuscript document.
Round 2
Reviewer 1 Report
Unfortunately, the author did not carefully consider the detailed comments on their first revision. Problems remained the same. The novelty is marginal. It seems this research focus on a local issue in Spain. However, for readers of international journals, the underlying theoretical contribution has not been highlighted. I do not think that it should be accepted for publication.
Author Response
REVIEWER’S COMMENTS
REVIEWER 1
Unfortunately, the author did not carefully consider the detailed comments on their first revision. Problems remained the same. The novelty is marginal. It seems this research focus on a local issue in Spain. However, for readers of international journals, the underlying theoretical contribution has not been highlighted. I do not think that it should be accepted for publication.
Response:
Dear reviewer,
First of all we apologize that we have not correctly highlighted the changes made in the manuscript according to your suggestions.
We are sorry for the mistake and then we express again in the response letter what we have done and tell you that we have highlighted them in red in the manuscript document that we have uploaded to the platform.
Thanks a lot
REVIEWER 1
- In 2.4, authors mentioned “through a review of the literature and based on the knowledge categories and clinical experience in the study pathology, the research team added a total of six of their own statements specific to dementia to the original instrument matrix”. It requires more literature to make a strong argument for six of their own statements.
Response 1: We appreciate this evaluation. We have added more information based on the literature in order to follow their recommendations.
- The procedure of the proposed method should be provided in a flowchart or diagram. The presentation of the results showed some fragility. How about presenting the results obtained on each step in the research on a board?
Response 2: Totally agree with your suggestion. From this, we have added a diagram in which the stages are specified
- The pre-final version of the EKSO-DEM/RKSO-DEM instrument in Spanish, consisting of 46 items, should be provided in the appendix.
Response 3: Thank you very much for your suggestion. We have added what you requested.
Reviewer 2 Report
In response to your previous point, I believe the introduction has been improved.
Please spell out the PwD (line 35) for the first time.
Please correct Table 1 and Table 5 as they are quite disorganized and confusing.
As for KESO-DEM/RKSO-DEM, maybe I am reading it wrong, but doesn't the original version consist of 6 factors for both? I checked a relatively old paper on this scale https://doi.org/10.1111/j.1365-2648.2007.04408.x and it shows both KE and RK as having 6 factors. However, in this study
(line 372) "The factor analysis with varimax rotation showed 7 factors with a value >1 for the KESO-DEM scale, and 6 factors with a value >1 for the RKSO-DEM (Table 3). ". I don't think there is any mention in the discussion of what was different about the factor structure or why the results were different. If it is different from the original factor structure, it must be mentioned. For example, was it the Spanish culture that caused the difference (what was the original culture and how did some aspect of the Spanish culture influence it, etc.)? Or, if it is a target characteristic, or a disease characteristic, etc., you will need to mention them.
Conclusion needs to be modified. The Conclusion needs to clearly state what you found out as a result of what you did with whom, supported by the data.
Author Response
REVIEWER 2
Response to reviewer
- Please spell out the PwD (line 35) for the first time.
Response 1: Thank you for your comments. We have corrected on the text.
- Please correct Table 1 and Table 5 as they are quite disorganized and confusing.
Response 2: Thank you for your observation. Tables have been modified.
- As for KESO-DEM/RKSO-DEM, maybe I am reading it wrong, but doesn't the original version consist of 6 factors for both? I checked a relatively old paper on this scale https://doi.org/10.1111/j.1365-2648.2007.04408.x and it shows both KE and RK as having 6 factors. However, in this study
(line 372) "The factor analysis with varimax rotation showed 7 factors with a value >1 for the KESO-DEM scale, and 6 factors with a value >1 for the RKSO-DEM (Table 3). ". I don't think there is any mention in the discussion of what was different about the factor structure or why the results were different. If it is different from the original factor structure, it must be mentioned. For example, was it the Spanish culture that caused the difference (what was the original culture and how did some aspect of the Spanish culture influence it, etc.)? Or, if it is a target characteristic, or a disease characteristic, etc., you will need to mention them.
Response 3: Comments have been added to the discussion reflecting these changes.
- Conclusion needs to be modified. The Conclusion needs to clearly state what you found out as a result of what you did with whom, supported by the data.
Response 4: Conclusion has been modified